# Gestational route to healthy birth (GaRBH): protocol for an Indian prospective cohort study

Vipin Gupta,[1] Ruchi Saxena,[2] Gagandeep Kaur Walia,[3] Tripti Agarwal,[3] Harsh Vats,[1] Warwick Dunn,[4] Caroline Relton,[5] Ulla Sovio,[6] Aris Papageorghiou,[7] George Davey Smith,[5] Rajesh Khadgawat,[8] Mohinder Pal Sachdeva[1]

VG, RS and GKW contributed equally.

For numbered affiliations see end of article.

**Correspondence to**
Dr Vipin Gupta;
udaiig@gmail.com

## ABSTRACT

**Introduction** Pregnancy is characterised by a high rate of metabolic shifts from early to late phases of gestation in order to meet the raised physiological and metabolic needs. This change in levels of metabolites is influenced by gestational weight gain (GWG), which is an important characteristic of healthy pregnancy. Inadequate/excessive GWG has short-term and long-term implications on maternal and child health. Exploration of gestational metabolism is required for understanding the quantitative changes in metabolite levels during the course of pregnancy. Therefore, our aim is to study trimester-specific variation in levels of metabolites in relation to GWG and its influence on fetal growth and newborn anthropometric traits at birth.

**Methods and analysis** A prospective longitudinal study is planned (start date: February 2018; end date: March 2023) on pregnant women that are being recruited in the first trimester and followed in subsequent trimesters and at the time of delivery (total 3 follow-ups). The study is being conducted in a hospital located in Bikaner district (66% rural population), Rajasthan, India. The estimated sample size is of 1000 mother-offspring pairs. Information on gynaecological and obstetric history, socioeconomic position, diet, physical activity, tobacco and alcohol consumption, depression, anthropometric measurements and blood samples is being collected for metabolic assays in each trimester using standardised methods. Mixed effects regression models will be used to assess the role of gestational weight in influencing metabolite levels in each trimester. The association of maternal levels of metabolites with fetal growth, offspring's weight and body composition at birth will be investigated using regression modelling.

**Ethics and dissemination** The study has been approved by the ethics committees of the Department of Anthropology, University of Delhi and Sardar Patel Medical College, Rajasthan. We are taking written informed consent after discussing the various aspects of the study with the participants in the local language.

## INTRODUCTION

The profound metabolic changes during healthy pregnancy ensure optimum fetal development and appropriate nutrient allocation between the mother and fetus.[1 2]

## Strength and limitations of this study

► This is a longitudinal study in India to examine trimester-specific metabolomic changes during the course of pregnancy and their influence on fetal growth.
► The study is collecting biological measurements on pregnant women during each trimester (first, second and third trimester) and their offspring at the time of birth.
► Newborns will be intensively phenotyped for the levels of metabolites and body composition.
► Unlike other Indian cohorts, the study participants are being recruited from a government hospital that caters for a rural population.
► Lack of information on prepregnancy measurements will restrict our analysis to the duration of pregnancy only.

Such metabolic adjustments are required for meeting raised physiological needs of pregnancy and for providing additional energy to support labour and lactation.[1] The concept of *'the developmental origins of health and disease'* suggests that exposures during critical periods in utero can lead to *'programmed'* structural and/or functional alterations that can transmit to the next generation and predispose them to disease at later age.[3 4] A study on 115 women with normal pregnancy has found that higher circulating fatty acids are associated with lower blood pressure and more favourable metabolic phenotype of the mothers.[5] The disparity in metabolic changes during pregnancy may influence fetal growth, and can increase the likelihood of adverse metabolic profiles across the life-course for both mother and the offspring through fetal programming.[6] For instance, increased transport of glucose and free fatty acids to the fetal-placental unit can lead to adverse fetal growth (ie, macrosomia).[7 8]

The biological system-wide understanding of metabolism requires us to capture

BMJ

phenotypic changes of the human *metabolome*, which comprises the full spectrum of metabolites present in all cells, tissues or fluids of the human body,[9–11] in different physiological states (ie, puberty, pregnancy, lactation and premenopause/postmenopause). The metabolome can be considered as an intermediate phenotypic mixture of metabolites encoded by our genome, influenced by specific environmental exposures and it shapes an individual's *metabolic phenotype or* metabotype.[10–12] It can be studied through the holistic approach of *metabolomics* that provides a quantitative fingerprint of all metabolites present in cells, tissues or an organism and can be measured using high-throughput technologies.[13 14] The field of metabolomics is important for uncovering quantitative metabolic trajectories of normal and abnormal pregnancies. The studies on quantitative estimation of longitudinal metabolomic changes during gestation are globally few in number[15–17] and there are none we know of in India.

## Metabolomic changes in pregnancy

The incomplete understanding of the complex nature of metabolic processes during pregnancy has generated considerable interest in exploring the effect of gestational characteristics on metabolomic changes during pregnancy and its impact on offspring health. Hyperglycaemia during pregnancy is one of the widely studied examples of pregnancy-induced metabolic stress which increases maternal risk of type 2 diabetes and also predisposes the developing fetus to poor metabolic health.[18] Studies have also assessed the metabolomics of pregnancy complications, such as, gestational diabetes,[19 20] placental abruption[21] and birth outcomes, such as preterm birth,[22] small for gestational age[23] and large for gestational age.[24]

An exploratory study has investigated the changes in metabolite levels during pregnancy among women with gestational diabetes in the third trimester and found elevated levels of branched chain amino acids, leucine, isoleucine and valine.[25] The catabolic nature of third trimester is represented by a decrease in maternal glucose levels and substantial rise in the levels of fasting insulin which in turn can lead to insulin resistance.[26] Moreover, circulating levels of lipids and lipoproteins rise twofold to threefold in late pregnancy.[27] Maternal amino acid levels change longitudinally due to placental uptake and fetal transfer from first to third trimesters.[15] There is an increase in energy production through the TCA cycle during the course of gestation along with concomitant decreases in free carnitine and acetylcarnitine and increase in carnitine palmitoyltransferase-1 activity and β-hydroxybutyrate levels.[15] A recent study has detected a role for metabolites in fetal sex differences in placental complications. It was found that higher maternal levels of spermine metabolite N1,N12-diacetylspermine (DiAcSpm), from the first to third trimesters, are observed in the female placenta and in the serum of women with a female fetus.[17]

## Relationships between maternal gestational weight gain and newborn health

Gestational weight gain (GWG) is a complex biological trait which is influenced by maternal physiology, maternal, placental and fetal metabolism.[28] There are short-term or long-term implications of inadequate or excessive GWG which direct the trade-off between maternal and child health outcomes.[28–32] For example, inadequate weight gain is associated with high risks of low birth weight, small for gestational age and preterm birth.[33] Excessive GWG is associated with large for gestational age, gestational diabetes, caesarean section and postpartum weight retention.[34] Furthermore, high prepregnancy maternal body mass index (BMI) is associated with perturbations to maternal metabolome across gestation.[35]

Maternal GWG is a good indicator of fetal nutrition.[36] Any amount of weight gain in the first 14 weeks and high rate of weight gain (>500 g/week) in 14–36 weeks are associated with raised adiposity in the offspring.[37] GWG in the first trimester is associated with offspring BMI from infancy to early childhood whereas excessive GWG (ie, ≥500 g/week) in second and third trimesters is associated with increased birth weight.[38] In the first half of the pregnancy (<20 weeks), excessive GWG is associated with higher birth weight and high body fat in newborns compared with the second half of pregnancy.[39] Furthermore, Gaillard *et al*[40] observed that early GWG was associated with increased adiposity levels and adverse cardiometabolic profile in childhood independent of prepregnancy weight. A study that investigated the effect of trimester-specific GWG on newborn anthropometry has found that women with an excessive rate of GWG in mid/late pregnancy deliver the heaviest babies.[41] Thus, the role of optimum GWG in fetal and child health is well studied[42] but the underlying biological mechanisms or mediating pathways are as yet unknown. The maternal fat deposition in early pregnancy due to excessive GWG may lead to maternal dysmetabolism[43 44] and effect the levels of maternal metabolites in subsequent trimesters. A study has found that the levels of triglycerides, high-density lipoprotein-cholesterol, apolipoprotein A1 and interleukin-6 are linearly associated with GWG across all levels of GWG in mid-pregnancy.[37] The authors postulated that mothers with high GWG may engage in lifestyles during and after pregnancy that may encourage weight gain. However, a few Indian studies have incorporated the GWG in their analyses, for example, an association of inadequate or excessive GWG with preterm deliveries,[45] average GWG among women with gestational diabetes[46] and maternal dietary intake of milk protein in first trimester on GWG.[47] But there is no Indian study that has explored the association between GWG and levels of metabolites. The continuous variability in the amount of weight gain during pregnancy, with respect to parity, height and geography,[48–50] has motivated the research for detecting the predictors and outcomes of GWG.

### Maternal metabolic associations with newborn phenotypes

Maternal prepregnancy obesity can predict higher branched chain amino acids and related metabolites in offspring.[51] The increase in maternal prepregnancy BMI is strongly associated with adverse cardiometabolic profile (very low-density lipoprotein [VLDL], VLDL-cholesterol, VLDL-triglycerides, VLDL-diameter, branched/aromatic amino acids, glycoprotein acetyls and triglycerides) in offspring.[52] A Mendelian randomisation study has suggested causal relationships between genetically elevated maternal prepregnancy BMI (1 SD) and blood glucose levels (7.2 mg/dL) with higher birth weight (55 g [95% CI: 93 g; p=0.004] and 114 g [95% CI: 147 g; p=7×10$^{-9}$], respectively) of the offspring.[53] The metabolic shift from first to third trimester of gestation can influence newborn phenotypes. A large study related to the effects of maternal metabolic signatures on fetal growth has identified urinary metabolites (like branched-chain amino acids, isoleucine, valine, leucine, alanine, choline, 3-hydroxyisobutyrate, progesterone and oestrogens) predictive of fetal growth, and explained 48%–53% of total variation in birth weight.[16] Recently, a pilot study has shown that both gestational age of the pregnancy (comparable to that estimated by ultrasound) and risk for preterm delivery can be predicted by measuring cell-free RNA transcripts in maternal blood during pregnancy.[54] This study highlighted the opportunities offered in the identification of non-invasive blood-based biomarkers of pregnancy outcome.[54] It has been shown that maternal hyperglycaemia is associated with higher cord blood concentrations of lysophosphatidylcholines in offspring independent of lifestyle intervention among obese women.[55] Thus, maternal prepregnancy BMI, RNA transcripts and hyperglycaemia during gestation influences fetal and newborn phenotypes.

Scholtens *et al*[56] assessed the impact of maternal hyperglycaemia, measured using an oral glucose tolerance test (OGTT), on fetal phenotypes at 28 weeks of gestation. They observed a decrease in amino acids, fatty acids and products of lipid metabolism, whereas a rise in triglycerides and carbohydrates was observed between fasting and 1 hour OGTT samples. Furthermore, the role of maternal glucose, triglycerides and fatty acids in predicting the newborn birth weight and sum of skinfolds thicknesses was also suggested. This confirmed the Freinkel modification of the Pedersen hypothesis, that is, fetal growth and fat accumulation in the setting of maternal hyperglycaemia is contributed to by other nutrients in addition to glucose.[57] The longitudinal assessment of the role of maternal metabolites during the length of the pregnancy on birth outcomes is restricted to four studies, of which two studies use urine samples[16 58] and the remaining two blood samples.[15 17]

### Gestational route to healthy birth cohort study

GWG in first trimester acts as an exposure for the levels of metabolites as an outcome in subsequent trimesters. Similarly, the levels of metabolites in a trimester can also

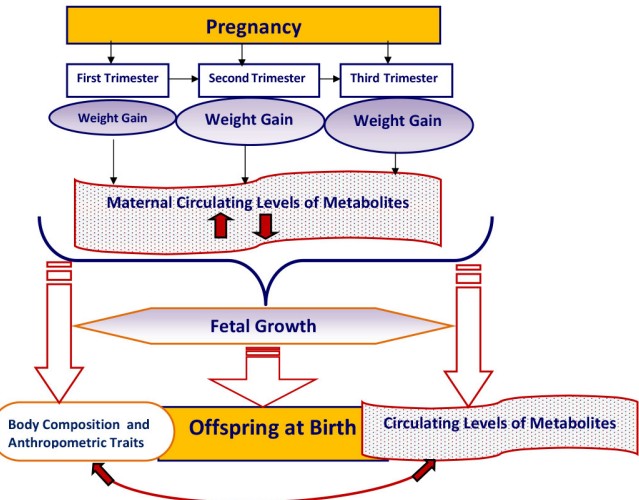

**Figure 1** Diagrammatic representation of the study hypothesis. We assume that the quantitative changes in maternal circulating metabolites are associated with gestational weight gain, and it influences fetal growth, offspring's levels of metabolites, and body composition and anthropometric traits at birth.

act as an exposure for GWG in next trimester(s). Therefore, this study aims to quantify differential metabolic variability in relation to maternal weight gain during pregnancy for assessing its influence, as an exposure, on a) fetal growth, b) offspring metabolite levels, c) body composition of offspring at birth. The conceptual model is given in figure 1. We are asking two research questions:

1. Are quantitative changes in levels of metabolites associated with GWG during pregnancy?
2. Are quantitative metabolic changes during pregnancy associated with fetal growth, offspring levels of metabolites and body composition at birth?

### Study objectives

► To detect differential changes in quantitative metabolites between the end of the first, second and third trimesters of pregnancy and to examine how these trajectories are associated with maternal weight gain.
► To determine the influence of metabolic changes during pregnancy on fetal growth, body composition (fat mass, lean mass and bone density) and anthropometry at birth (ie, body length, weight and circumferences) independent of maternal prepregnancy BMI and paternal phenotypic characteristics.
► To assess the association between maternal metabolic profile during pregnancy and metabolite levels of offspring at birth (in cord blood).

### Study area, design and setting
#### Study area

This study is being conducted in the government hospital of Sardar Patel Medical College, that is, 'Prince Bijay Singh Memorial Hospital' (PBM), located in the subdistrict *Bikaner*, which is one of the eight subdistricts ('*Tehsils*') of Bikaner district in Rajasthan state. According

to Census of India,[59] Rajasthan is India's largest state in terms of geographical area, with a population size of 68 million. The population of Bikaner district is 2.3 million (66% is rural) of which 47% are females. In Bikaner, crude birth rate is 26 per 1000 individuals, the total fertility rate is 3 and the infant mortality rate is 60 per 1000 individuals.[60] The majority of study participants are homemakers (80%), follow the Hindu religion (70%) and approximately 20% of the participants are illiterate. The choice of PBM hospital was based on the availability of government medical facility, which is the largest in district Bikaner and it ensures health care accessibility and affordability (due to free of cost treatment) for the entire population.

### Study design

A hospital-based cohort study which is recruiting and longitudinally following the pregnant women until delivery, and their offspring at birth in district Bikaner, Rajasthan, India.

### Sample size

Studies of fairly large size for detecting associations between longitudinal changes in levels of metabolites in pregnancy and fetal growth, and levels of metabolites of the offspring at birth, are few in number. For example, Ciborowski et al[61] used 770 samples collected during the first trimester for studying the relationship between maternal metabolic levels with macrosomia and low birth weight. Maitre et al[58] investigated 34 metabolites on 438 samples recruited during the first trimester and found an association with fetal growth restriction and preterm birth. The majority of studies related to metabolic changes during pregnancy[62–64] and maternal-fetal medicine[65 66] were based on 30–60 samples.[67 68] Exact power calculations are challenging due to a restricted number of prior studies leading to unknown features.[69] Even the largest study on 4212 participants for studying the predictive metabolomic biomarker for placental complications included limited sample of pre-eclampsia (n=134), fetal growth restriction (n=162) and controls (n=259).[17] Thus, the proposed sample size must be larger than the majority of informative studies conducted to date and feasible within the cost boundaries for the technology being used.

Tynkkynen et al[70] studied quantitative urine and serum metabolite data on 995 samples of Northern Finland Birth Cohort for establishing the proof of concept for conducting large-scale epidemiological studies in metabolomics using an example of BMI. A relatively large study (n=800) has found that maternal urine metabolites can independently explain 12% of birth weight after adjusting for all other lifestyle/clinical factors.[16] They also found an association of trimester-specific metabolic changes on fetal weight (change of 1.2% in first trimester and 2.4% in third trimester). Studies have observed that a metabolite level may change from 10% to 30% (even more for some metabolites) from first trimester to third trimester.[15 71]

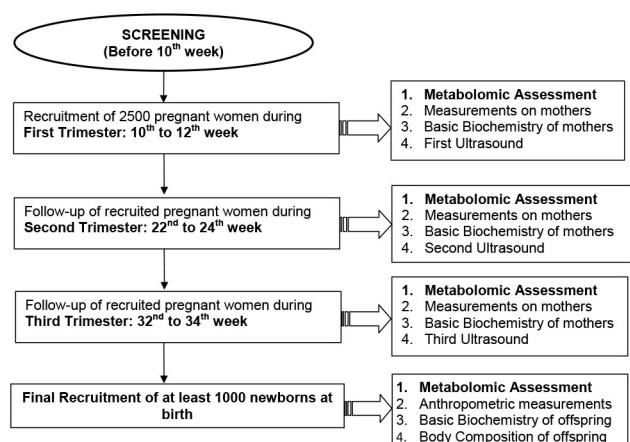

**Figure 2** Recruitment and follow-up strategy being used in GaRBH cohort.

For a cross-sectional analysis, a sample size of 1000 will be sufficient (with 98% power) to explain 4%–5% variation in the level of a metabolite by maternal weight as a predictor, even after correcting for multiple testing ($\alpha$= 0.0001; for number of metabolites). Moreover, a sample size of 1000 will also be sufficient to explain 2% variation in the fetal growth parameters (with 80% power, $\alpha$=0.0001) by maternal level of metabolite as a predictor.

Therefore, we propose to recruit 2500 pregnant women during the first trimester in order to measure metabolites on at least 1000 pregnant women (we may recruit >1000 women) in each trimester and their offspring, accounting for loss to follow-up.

### Recruitment strategy

Women visiting the antenatal clinic of the PBM hospital are being screened with the help of a one-page questionnaire to assess their eligibility for the study. We aim to screen around 6000 pregnant women from PBM hospital to reach the targeted sample size. The participants will have to satisfy the eligibility criteria (given below) before being recruited at baseline (figure 2) from February 2018 to March 2023:
1. 10 weeks of gestation;
2. Age: ≥18 to <35 years;
3. Singleton pregnancy;
4. Natural conception (ie, without any fertility treatment);
5. No long-term medication (ie, not more than 2 weeks in last 1 year);
6. No more than one miscarriage in previous pregnancies. In cases where the current pregnancy is the second pregnancy, then the women with miscarriage in their first pregnancy will also be excluded;
7. No previous pregnancy affected by pre-eclampsia/eclampsia;
8. No clinical evidence of sexually transmitted disease;
9. No hormonal therapy.

We will not exclude the study participants (recruited in first trimester) who develop pregnancy complications

in subsequent trimesters that may influence fetal growth (like pre-eclampsia,[72] eclampsia, intrauterine growth restriction and gestational diabetes mellitus[73]). Exclusion of these complications in current pregnancy would lead to a reduction in the natural variation in the study population. Furthermore, we will not exclude preterm delivery cases due to its relationship with fetal growth[74] and its high burden (ie, 12.8 million in 2010) in India,[75] as it could lead to bias by restricting the sampling to very healthy pregnancies which may not represent the general population.

## Patient and public involvement

The fieldworkers are trained to establish rapport with the study population, which begins with describing the research questions and objectives of the study to the pregnant women visiting the antenatal clinic of PBM hospital during first trimester. Once they agree to consider participation, project staff discusses the participant information sheet, containing detailed information on all the measures and outcomes of the project with them in local language. Our data collection process starts only after taking the informed written consent from the eligible participants. We constantly try to involve them during the pregnancy by providing the readings of biochemical parameters and ultrasounds in each trimester and facilitate their antenatal visits in PBM hospital. The project staff will also provide support to the participant during and after the childbirth in the hospital. Moreover, the study population will be engaged through public events for the final dissemination of the study results.

## ASSESSMENT PROCEDURES AND INSTRUMENTATION
### Ultrasound measures

For ultrasound-based measurements relating to fetal growth, we are using the existing standard operating procedures described by INTERGROWTH study[76] to ensure reliable data collection. The data are being collected by trained sonographers in Bikaner, who are highly familiar with the equipment (SonoACE R7 Samsung Healthcare) and experienced in conducting ultrasound examinations. The intraobserver variation will be assessed by repeat sonographic measurements on 10 participants on the same day. Comparisons will also be made between the readings of two sonographers of the hospital on 10 participants to assess interobserver variability. The interequipment variation in ultrasound measures will be assessed by comparing the PBM results of 10 participants with the radiology laboratory outside the hospital. We will measure the crown-rump length (CRL), biparietal diameter, occipitofrontal diameter, head circumference, femur length, anterior-posterior abdominal diameter. Gestational age based on CRL and fetal weight will be estimated with the help of INTERGROWTH equations.[76 77]

## Estimation of gestational age

The accurate estimation of individual-level gestational age during pregnancy is vital for interpreting fetal anatomy and growth patterns and predicting the date of delivery. Gestational age is being calculated from the first day of the last menstrual period (LMP). Moreover, for women providing clear information on LMP (in completed weeks and days) their gestational age is being confirmed with an ultrasound-based measurement of CRL in the first trimester, using an internationally recognised chart.[76] If the difference between the CRL and LMP estimates is ≤7 days, LMP is being considered valid and taken as the true biological date.[76] In women with an LMP and CRL difference of >7 days, CRL is being considered valid for estimating the gestational age.[78]

## Questionnaires

A comprehensive questionnaire has been developed for collecting information on demographic, anthropological (caste, religion and consanguinity), medical disease history, tobacco and alcohol intake and passive smoking. The validated food frequency questionnaire of '*Indian Migration Study*' is being used for evaluating dietary intake.[79] Maternal physical activity during pregnancy is being assessed using the international physical activity questionnaire (ie, 7 days long).[80–82] The assets-based tool of National Family Health Survey-4 is being used to evaluate socioeconomic position.[83] For assessing depression, we are using the Patient Health Questionnaire[84] and Edinburgh Postnatal Depression scale.[85] The WHO-5 questionnaire is being used for collecting information on quality of life.[86] The capability well-being, that is, individual's freedom to function in five key areas of their life: 'stability', 'attachment', 'achievement', 'autonomy' and 'enjoyment', is being assessed by using the ICEpop CAPability measure for Adults.[87–89] Information on maternal environment, gynaecological and obstetrics history is also being collected using tools adapted from INTERGROWTH study.[90 91] Additionally, a structured questionnaire on social factors like relationships with family members, husband position in household, social support during pregnancy and travel is being used for assessing the quality of social life and adequacy of social support (table 1).

## Adult anthropometry

The assessment of anthropometric traits (weight, height, sitting height, circumferences and skinfolds) are being performed on study participants using standard procedures.[92]

Weight is being measured when the participant standing straight on the centre of the weighing machine (Tanita, model-HD380), arms hanging freely with minimum clothes and without accessories. Height is being measured as the vertical distance between vertex (highest point on head) and floor when the participant standing straight with feet together on the level floor without footwear, arms resting by the side, with head in

**Table 1**  Measures, tools and time points of data collection

| | Mother | | | Offspring |
|---|---|---|---|---|
| | **First trimester** | **Second trimester** | **Third trimester** | |
| Timing of data collection | 10th to 13th weeks | 22nd to 24th weeks | 32nd to 34th weeks | Birth |
| Gynaecological and obstetric | ✔ | ✔ | ✔ | – |
| Socioeconomic position | ✔ | ✔ | ✔ | – |
| Anthropological (caste and religion) | ✔ | ✔ | ✔ | – |
| Education | ✔ | ✔ | ✔ | – |
| Marital status | ✔ | ✔ | ✔ | – |
| Smoking | ✔ | ✔ | ✔ | – |
| Alcohol drinking | ✔ | ✔ | ✔ | – |
| Medication | ✔ | ✔ | ✔ | – |
| Disease history | ✔ | ✔ | ✔ | – |
| Maternal environment | ✔ | ✔ | ✔ | – |
| Physical activity (IPAQ) | ✔ | ✔ | ✔ | – |
| Time-activity pattern | ✔ | ✔ | ✔ | – |
| Food frequency | ✔ | ✔ | ✔ | – |
| Dietary recall (3 days) | ✔ | ✔ | ✔ | – |
| PHQ-12 | ✔ | ✔ | ✔ | – |
| Well-being (WHO-5) | ✔ | ✔ | ✔ | – |
| Depression (EPNDS) | ✔ | ✔ | ✔ | – |
| Social factors | ✔ | ✔ | ✔ | – |
| Capability well-being (ICECAP-A) | ✔ | ✔ | ✔ | – |
| Anthropometry | | | | |
| Height/body length | ✔ | ✔ | ✔ | ✔ |
| Sitting height | ✔ | ✔ | ✔ | – |
| Weight | ✔ | ✔ | ✔ | ✔ |
| Circumferences | ✔ | ✔ | ✔ | ✔ |
| Skinfolds | ✔ | ✔ | ✔ | ✔ |
| Blood pressure | ✔ | ✔ | ✔ | – |
| Laboratory tests: Glucose, insulin, Hb, lipids, LFT and KFT and blood group | ✔ | ✔ | ✔ | ✔ |
| Metabolomic data | ✔ | ✔ | ✔ | ✔ |
| Ultrasound measures: CRL, BPD, OFD, HC, FL, AC and APAD | ✔ | ✔ | ✔ | – |
| Placenta and cord measures and images | – | – | – | ✔ |
| Body composition (DXA scan) | – | – | – | ✔ |

APAD, anterior-posterior abdominal diameter; BPD, biparietal diameter; CRL, crown-rump length; EPNDS, Edinburgh Post Natal Depression Scale; FL, femur length; HC, head circumference; ICECAP-A, ICEpop CAPability measure for Adults; IPAQ, International Physical Activity Questionnaire; KFT, kidney function test; LFT, liver function test; OFD, occipitofrontal diameter; PHQ, Psychological health Questionnaire.

eye-ear plane (Frankfort horizontal plane) and heels, buttocks and shoulders projected on the vertical plane using a wall-mounted stadiometer (ADE, Germany, model-MZ10023-3). Sitting height is being measured as the maximum distance from the vertex (top of the head) to the base of a flat sitting surface, with legs hanging freely and head in eye-ear plane.

Circumferences are being measured at the specific landmarks when the plane of the tape (ADE, model-MZ10021) is perpendicular to the long axis of the body and parallel to

the floor without tape being pulled too loose or too tight. Waist circumference is being measured at the narrowest point (which is an approximate midpoint) around the waist between the lower rib margin of the last palpable rib and the top of the iliac crest with the help of a measuring tape. Mid-upper arm circumference is being measured at the midpoint between the tip of the shoulder and the tip of the elbow (acromion and olecranon process, respectively) taken horizontally (at the middle of the biceps). Hip circumference is being measured at the widest part of the hip (buttock) at the level of the trochanterion, which is the most superior point on the greater trochanter of the femur, not the most lateral point. Calf circumference is being measured on the right leg as the maximum circumference in a plane perpendicular to the long axis of the calf until the greatest circumference is located when foot is placed on raised platform.

Skinfold measurements are being taken in relaxing muscles by grasping a distinct fold that separates from underlying muscles with the help of Harpenden Skinfold calliper when the jaws of the calliper are perpendicular to the length of the fold on the site of measurement. Skinfold measurements at biceps, triceps, subscapular, suprailiac, medial calf and anterior thigh are being taken to the nearest 0.1 mm.

Three readings are being taken for all the anthropometric measures with minimum acceptable difference of 100 g, 5 mm, 5 mm and 2 mm between the readings for weight, height, circumferences and skinfolds, respectively. The readings are being recorded to the last completed grams or millimetre.

### Infant anthropometry

We will use the protocol of the International Fetal and Newborn Growth Consortium[93] for measuring the infants. Birth weight will be taken within 30 min of delivery, without clothes and in stable condition when the infant is placed an electronic weighing scale (ADE, model-M114600) to the nearest 5 g. Infant length will be measured within 12 hours of birth using the baby length measuring board (ADE range 100–1000 mm), which has a fixed headboard and moveable footboard, placed on a raised flat platform on the level surface. The head of the infant will be positioned in eye-ear plane on the board and holding the infant's legs with the left hand and moving the footboard with the right hand. Head circumference will be measured when the infant is on the lap of the field investigator, with stable head, using a measuring tape (SECA, 212) anchored just above the eyebrows on the forehead, and at the back the tape being positioned over the fullest protuberance of the skull. Three readings will be taken for all the newborn anthropometric traits with minimum acceptable difference of 10 g, 5 mm and 2 mm between the readings for weight, infant length and circumferences, respectively. The readings will be recorded to the last completed millimetre.

All the equipment used for the assessment of adult and infant anthropometric traits is being calibrated twice a week with the help of standard weights and measures.

### Blood pressure

Maternal blood pressure is being measured in the sitting position during each trimester after a rest of 5 min before taking the first reading, using an Omron blood pressure monitor (model: HBP-1300) validated for pregnant women[94] and cuff of appropriate size. Three readings will be taken with 2 min interval between the readings with maximum acceptable difference of 5 mm Hg between two readings. The equipment used is being calibrated thrice a week with the readings of mercury sphygmomanometer.

### Biological sample collection and storage

A 15 mL of blood sample is being collected during each trimester from the study participants after overnight fasting (10 hours). We are using EDTA-coated, sodium fluoride-coated and plain tubes for blood collection for the separation of EDTA plasma, fluoride plasma and serum samples, respectively. Serum will be used for determining the levels of metabolites in addition to triglycerides, cholesterol, lipids, kidney function test (KFT), liver function tests (LFT) and insulin, whereas fluoride plasma will be used for measuring glucose levels. EDTA tube is centrifuged at 3500 rpm for separating the white blood cells (WBCs) (buffy coat), plasma and red blood cells. After separating the plasma, EDTA tube containing WBCs will be stored for future extraction of DNA for genetic and epigenetic studies. Multiple aliquots of serum and plasma collected at each trimester are being stored in a −80°C freezer (table 2).

We will collect cord blood and placenta at the time of delivery (table 2) and processed with the help of the standard operating procedure used in the 'Avon Longitudinal Study of Parents and Children', University of Bristol, UK. The cord blood will be collected in EDTA-coated, heparin-coated and plain tubes (10 mL each) and these will be processed for the separation of plasma and serum. The remaining sample in EDTA tube with buffy coat will be stored in a freezer at −80°C. In order to stabilise the gene expression, RNA*later* will be added in heparin tube. Placenta will be processed immediately after delivery and aliquots of tissue sample from maternal and fetal sides, cord tissue and membrane will be stored in a freezers at −80°C (table 2). All the stored aliquots will be appropriately bar-coded, stored independently in a separate storage box.

### Laboratory tests

The fasting serum samples will be transferred to the Department of Anthropology, University of Delhi, on dry ice for the generation of data on biochemical assays on all the mothers during each trimester and also on their offspring using cord blood. The available auto-analyzer (XL-640, ERBA Mannheim, Czech Republic) in the Department of Anthropology, University of Delhi,

**Table 2** Type and number of aliquots of biological samples

| | Pregnancy | | | Offspring |
|---|---|---|---|---|
| | **First trimester** | **Second trimester** | **Third trimester** | |
| Blood samples | | | | |
| Serum | Six aliquots | Six aliquots | Six aliquots | – |
| EDTA plasma | Two aliquots | Two aliquots | Two aliquots | – |
| Fluoride plasma | Two aliquots | Two aliquots | Two aliquots | – |
| Buffy coat | One aliquot | One aliquot | One aliquot | – |
| Cord blood samples | | | | |
| Serum | – | – | – | Four aliquots |
| EDTA plasma | – | – | – | Three aliquots |
| Heparin plasma | – | – | – | Three aliquots |
| Buffy coat with RNA*later* | – | – | – | One aliquot |
| Buffy coat | – | – | – | One aliquot |
| Placenta samples | | | | |
| Maternal side tissue | – | – | – | Eight aliquots |
| Maternal side tissue with RNA*later* | – | – | – | Four aliquots |
| Fetal side tissue | – | – | – | One aliquot |
| Fetal side tissue with RNA*later* | – | – | – | One aliquot |
| Membrane tissue | – | – | – | One aliquot |
| Cord tissue | – | – | – | One aliquot |
| Cord tissue with RNA*later* | – | – | – | One aliquot |

India, will be used for biochemistry data generation (ie, glucose, insulin, lipids, KFT and LFT) after ensuring daily calibration. Insulin levels will be estimated using ELISA kits. Furthermore, a reagent lot calibration will also be performed whenever there is a change in lot number of the reagents to estimate the variation in assays between two reagent lots. For internal quality control, one blank per 100 samples and one control sample from *UK National External Quality Assessment Service* of known concentrations will be run in every 200 samples. Moreover, one duplicate and one sample of known concentration from the previous batch will be run per 100 samples for assessing intra-assay and inter-assay variation, respectively, to help with analyses in confirming repeatability and reproducibility of biochemistry data. Finally, 5% of the samples will be run in a different laboratory for all the biochemical assays as a measure of external quality control. Moreover, information on participant's blood group, Sickle hemoglobin (HbS) and HIV infection, which are the part of standard protocol of the hospital's antenatal care, will be taken from PBM Hospital.

## Quantitative metabolomics
The quantitative data on the level of metabolites (in mothers during each trimester and in offspring using cord blood) will be generated through a targeted approach for screening the predefined and known metabolites using a commercial assay, that is, AbsoluteIDQ p400 HR kit (Biocrates Life Sciences, Innsbruck, Austria). It provides data on 400 endogenous metabolites from 11 metabolite classes (ie, amino acids, biogenic amines, monosaccharides, acylcarnitines, diglycerides, triglycerides, lysophosphatidylcholines, phosphatidylcholines, sphingomyelins, ceramides and cholesteryl esters). The choice of AbsoluteIDQ p400 HR kit based on proven utility of its previous version of kit (AbsoluteIDQ p180) in several metabolomics-based epidemiological studies.[9 95–98] Our collaborator, the Phenome Centre Birmingham, University of Birmingham, UK, has experience of using this kit in an interlaboratory trial. For generating data on targeted metabolites, a triple quadrupole mass spectrometry-based technology will be used which characterises a metabolite by its molecular mass, its specific fractionation pattern (tandem mass spectrometry) and high sensitivity.[99] The complete analytical process will be performed using the MetIQ software package, which is an integral part of these kits. The quality control procedure used by Suhre *et al*[100] will be followed for generation and cleaning of metabolomic data. One reference sample will be measured three times on each plate across the samples. The quality of metabolite data will be controlled by estimating their coefficient of variation and missing value rates.[15] Outlying samples and data points will be excluded and finally all missing values will be imputed.[101]

## Body composition
The body composition of infants will be measured using a DXA machine (Osteo Pro Max, BM Technology, Korea)

available in PBM Hospital, Bikaner. The machine will be calibrated using infant phantoms at regular intervals. We have developed a standard operating procedure with the help of published literature[102–104] for neonatal scans of femur, spine and whole body. Newborns will be measured within 48 hours of delivery after feeding in a calm or sleeping state. Infants will be swaddled in a thin cotton blanket in the required position and measurement will be taken without any metals, buttons or pins on the infant. The region of interest (ROI) for anterior-posterior spine will be five vertebral bodies of the lumber region; for femur scan this will include the femoral neck, trochanters, intertrochanteric and total hip regions. For the whole body scan, the infant will be positioned at the centre of the table, aligned in a straight line from the nose, centre of the body and down through the knees and toes. Six ROIs will be taken for the whole body scan (head, entire right and left arms, right and left leg regions and trunk).

### Data capture, protection and management
The data are being recorded in paper-based questionnaires that will be initially entered in Microsoft Excel, and later transferred to a SQL-based database to be developed in collaboration with the University of Bristol, UK. The master data will be stored at multiple locations (at least two) and managed in computers available in the Department of Anthropology, University of Delhi. One unique identification number (UID) is being assigned to each study participant (mothers) that will be common for her questionnaire-based information, biological measurements, blood and other biological samples. Similarly, newborns will also be assigned a UID which is linked to the UID of mothers. The computers will be password protected, and UID linked with personal identifiers necessary for contact and future follow up, will be encrypted and held separately in a different system and backed up regularly. All the biological samples will be collected in pre-assigned bar-coded tubes and vials and will be stored at −80°C in Bikaner, Rajasthan. A comprehensive data dictionary will be prepared for all the study variables and the entire data will be regularly checked for outliers, distribution of variables and missingness.

### Data analyses
The evaluation of cross-sectional associations of maternal body weight with quantitative metabolite levels in each trimester, independently, will be done using regression-based modelling. Both level of metabolites and gestational weight in each trimester will act as exposure as well as outcome. Variables like socioeconomic position, gestational age, parity, diet, smoking and physical activity will be considered as confounders. Maternal characteristics will be considered as exposures (like trimester-specific weight, metabolite levels) to evaluate their association with offspring-related outcomes (fetal growth, birth weight, body compositions and metabolite levels). The potential confounders for maternal-offspring associations will be socioeconomic position, gestational age, parity, diet, sex of the offspring, smoking and physical activity.

Mixed effects regression models for repeated measures will be applied for examining longitudinal associations of weight gain from first to third trimester[42] with maternal levels of metabolites during pregnancy and fetal growth.[105] We will also analyse the associations of level of metabolites from first to third trimester with trajectories of GWG. Mediation analyses will be conducted for evaluating the role of maternal metabolites as intermediates between GWG and offspring's body composition.[106] False discovery rate based multiple testing correction for number of metabolites will be applied.[107] Mediation analysis will be conducted to examine whether the effect of GWG on fetal growth is mediated through alteration in levels of circulating metabolites.

### Public health implications
In India, higher prevalence of both malnutrition and overnutrition,[108 109] increased rates of gestational diabetes[110] and pre-eclampsia[111] are common challenges related to maternal-fetal health. Moreover, India has high rates of birth defects, that is, over 60 per 1000 live births,[112] higher number of infants born small for gestational age,[75] still births[113] and >45% of children under 5 years of age are stunted,[114] which need to be addressed through translational research. The metabolomic approach will enable the identification of biochemical markers predictive for undesirable birth outcomes.

The role of the intrauterine environment, through fetal overnutrition, has been proposed in considering determination of higher birth weight, childhood obesity, impaired glucose tolerance and dyslipidemia.[115–117] The knowledge of trimester-specific quantitative levels of metabolites will uncover mechanistic pathways relevant for optimum fetal growth and healthy birth outcomes. Understanding of changes in gestational metabolism will provide many opportunities for detecting potentially relevant metabolites for planning effective interventions during pregnancy.

### Ethics and dissemination
We are taking written informed consent after discussing the various aspects of the study with the participants in the local language. The findings of the study will be disseminated through peer-reviewed national and international journals and through conferences and seminars.

**Author affiliations**
[1]Department of Anthropology, University of Delhi, Delhi, India
[2]Department of Obstetrics and Gynaecology, Sardar Patel Medical College, Bikaner, Rajasthan, India
[3]Public Health Foundation of India, New Delhi, India
[4]School of Biosciences, Phenome Centre Birmingham and Institute of Metabolism and Systems Research, University of Birmingham, Birmingham, UK
[5]MRC Integrative Epidemiology Unit and Bristol Medical School, University of Bristol, Bristol, UK
[6]Obstetrics and Gynecology, University of Cambridge, Cambridge, UK
[7]Nuffield Department of Women's & Reproductive Health, University of Oxford, Oxford, United Kingdom

[8]All India Institute of Medical Sciences, New Delhi, India

**Acknowledgements**  The authors would like to thank the study participants, project's field investigators, medical staff and administration of PBM hospital, Bikaner, Rajasthan for facilitating the study.

**Contributors**  VG conceived the study concept with the help of GDS. VG developed the funding proposal, applied for funding and initiated the writing of the protocol paper. VG, GKW and MPS developed the study tools and procedures. VG, RS, RK, MPS and GKW coordinated the implementation and running of the study. GDS has advised in developing the study protocols and quality control and assurances. WD has provided expert opinion on metabolomic data generation and quality control. AP has provided expert opinion on ultrasound measures. GDS, CR, US, WD and VG have contributed to the statistical analysis plan and data interpretation. VG, RS, GKW, RK, TA and HV contributed to data collection and field management. All authors contributed in the revision and approval of the final version of the manuscript.

**Funding**  This study has been supported by Wellcome/DBT India Alliance, Hyderabad, India, 'Intermediate' fellowship award to Vipin Gupta (grant reference: IA/CPHI/16/1/502623). ATP is supported by the Oxford Partnership Comprehensive Biomedical Research Centre with funding from the NIHR Biomedical Research Centres funding scheme.

**Competing interests**  None declared.

**Patient consent for publication**  Not required.

**Ethics approval**  The study has been approved by the ethics committees of the Department of Anthropology, University of Delhi and Sardar Patel Medical College, Bikaner, Rajasthan.

**Provenance and peer review**  Not commissioned; externally peer reviewed.

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
