## [Reviewer comments · BMJ Open]

ARTICLE DETAILS

TITLE (PROVISIONAL)	Gestational Route to Healthy Birth (GaRBH): protocol for an Indian prospective cohort study
AUTHORS	Gupta, Vipin; Saxena, Ruchi; Walia, Gagandeep; Agarwal, Tripti; Vats, Harsh; Dunn, Warwick; Relton, Caroline; Sovio, Ulla; Papageorghiou, Aris; Davey Smith, George; Khadgawat, Rajesh; Sachdeva, Mohinder

VERSION 1 - REVIEW

REVIEWER	Xiaobin Wang Johns Hopkins University USA
REVIEW RETURNED	05-Aug-2018

GENERAL COMMENTS	Title: Gestational Route to Healthy Birth (GaRBH): a prospective cohort protocol Summary: This protocol describes a prospective study aiming to recruit and follow about 1,000 mother-offspring pairs to examine the association of maternal metabolites at each trimester with fetal metabolites at birth as well as maternal gestational weight gain and fetal growth and newborn anthropometric traits at birth. Overall, the protocol is well-written. The scientific premise of this study is strong, and the research questions are important. The study design is appropriate for stated objectives. Data collection procedures and quality assurance are described in detail. Specific comments: 1. Study participants This study is being conducted in a hospital located in Bikaner district (66% rural population), Rajasthan, India. It would be helpful to provide information on the socio-demographic characteristics of the pregnant mothers attending this hospital, including educational levels.2. Ethics Please clarify if all the study participants have adequate literacy level to give written informed consent? Minor comments:
--

	1. Estimation of Gestational Age: On page 14 of 29, line 13, "is longer than <=7 days", looks like a typo? 2. On page 12 of 29, line 17, "being calibrated thrice a week", should be "twice"
--	---

REVIEWER	Alexander Manu Liverpool School of Tropical Medicine, UK
REVIEW RETURNED	21-Sep-2018

GENERAL COMMENTS	General comments: This is a relevant study that has substantial potential to contribute to a better understanding of the biological mechanisms underlying various adverse pregnancy and birth outcomes. The study of the metabolomics is becoming increasingly popular in global science and the findings may have very wide applicability in low- and middle-income settings around the globe, where majority of these adverse pregnancy and birth outcomes occur. However, these are my comments on the protocol  1. The order of presentation of the study procedures does not make easy reading. For instance, description of the quantitative analysis of the metabolomics is made before how and what samples are being collected and processed. Data collection is described before stating that the gestational age will be confirmed with ultrasound. 2. The manuscript could be made more precise and concise. There are too many repetitions and lots of material that do not add to the value of the research. It may be the reason why there are 113 references. Many of these can be taken out without affecting the relevance and quality of the study. 3. The introduction does not make a case for the current gaps in knowledge which this study seeks to address 4. There are some methodological and procedural imperatives that will need to be looked at carefully. For instance, the procedure for measuring MUAC; the decision to use Omron sphygms without stating whether they are calibrated for use in pregnancy; the choice of LMP over ultrasound for early pregnancies where the two differ; etc. 5. Sentence construction is very complex and sometimes the use of different tenses makes reading a little difficult. [ ] Title of the study: The title of the study may be slightly misleading. It is not clear what gestational route to healthy birth means. Every birth follows the gestational route irrespective of the outcomes – healthy or unhealthy birth. The study does not state that it is seeking to understand how to improve the gestational route in order to ensure a healthy birth. Moreover, I will suggest that the authors clearly indicate that this is the "prospective cohort study" rather than "prospective cohort protocol". In my estimation, the two are not the same. [ ] Abstract
---

In the abstract, the authors have made the for how important this study could potentially be. However, the import of a protocol is to state how the study is going to be done. There is nothing on what biological samples (blood or urine) are going to be collected, when and how these are going to be processed to ensure that the research questions can be answered.

Also, in the introduction the aim of the study is not clear as to whether the GWG is thought to depend on the variation in the level of the metabolites or the level of the metabolites will depend on GWG when the authors used the phrase "...in relation to GWG and its influence on...", whereas in the methods, the analysis seems to be clearly looking at "the role of gestational weight in influencing metabolite levels in each trimester and the association of maternal levels of metabolites with foetal growth, offspring's weight and body composition at birth"

Even in the latter, the word "influencing" is scientifically unclear. Are they measuring association between GWG and metabolite levels and how these are, in turn, related to fetal growth and newborn anthropometric measurements? This needs to be clear.

Study strengths and limitations

- What do the authors mean by "collecting objective measurements on pregnant women ..."?
- So far, the study objectives seem to suggest that the metabolites will be assessed in the mother in relation to the "influence" of their levels on fetal growth. Are the authors suggesting, in bullet 3, that metabolite levels in newborns will also be assayed?
- In bullet 4, the authors are exaggerating the strength of the study by claiming that most Indian cohorts are NOT from rural areas. Meanwhile they do not mention the obvious selection bias inherent in the fact that the study is entirely hospital-based. In the rural population, this is significant since many do not even have access to hospitals in order to be part of this study. Either the very ill mothers or the (relatively) richer population who can afford will end up being in hospital and hence will be recruited. This is a limitation which the authors need to mention.

Introduction

- There are repetitions in the text that when avoided could have made the manuscript much shorter. For instance, pg5L5 and pg5L13 are both talking about the essence of the metabolic changes in pregnancy. The first paragraph therefore needs revision so that it becomes a bit more precise and concise.
- The authors, on pg5L16, cited a study to support their claim on how metabolic changes promote fetal development but the said study found that increased FFAs only lowered maternal blood pressure and produced "more favourable metabolic phenotype for the mother". The latter is quite vague and there was nothing about fetal development.
- On pg5 paragraph2, it is not clear what the authors mean by system-wide understanding of metabolism. Is it "system-wide approach to understanding human metabolome"?

On L37 of para2, did you mean cells, tissues and organs?

The last sentence of para2 does not read well. Am I right in thinking that the authors wanted to say that “only few longitudinal studies have quantitatively estimated the changes in the level of metabolomes at the different stages of pregnancy”? If so, the current construction needs revision.

The two paragraphs under “Metabolomic changes in pregnancy” seem to contradict the last sentence just before that because whilst the authors claim studies quantifying metabolomics are few, in the 2 paragraphs alone, they have cited about 10 different studies. Could the authors also provide a chronological account of the cited changes during pregnancy from the first to the third trimester so that the picture becomes clearer to the reader?

The section on “Relationships between maternal gestational weight gain and newborn health” has many very good points but will need to be re-written for the message to be clearer. As it is now, it reads as though the authors are presenting facts as they found them without guiding the reader to a clear message. It requires a bit of synthesis and focus.

In the section on “Maternal metabolic associations with newborn phenotypes”, again, the authors are clear with the message. The 3rd sentence on the Mendelian randomisation study could have stated the main findings i.e. causal associations between maternal BMI, blood glucose levels in pregnancy and infant birthweight. They could then have gone on to provide the exact quantification of the association and indicate statistical significance of the association. Also there was the mention of 10 metabolites but no identities of these metabolites are provided leaving the reader wondering what they might be. The authors on pg 7 of 29 L26 re cfRNA also refer to “maternal gestational age” when I think they were referring to the gestational age of a pregnancy.

Under the GaRBH cohort, I think you missed out “study” in the heading because you were not describing the cohort but the purpose of the study. Meanwhile, there is no clear rationale for the study because the previous paragraphs in the introduction have not clearly identified the current gaps in knowledge that the authors aim to address with this study. The question the reader will ask is “What is not known about this topic?”. The research questions, therefore, seem to have no rationale.

Study Objectives

The first objective starts with an assumption that the changes are going to be different in the different stages of pregnancy. This could be a subject of investigation under GaRBH. It could have been restated as “to determine whether the quantitative levels of metabolites differ at different stages of pregnancy and whether the trajectories are associated with maternal GWG”

The second objective should clearly state indicate that the associations will be measured independent of maternal pre-pregnancy BMI and paternal phenotypic characteristics. I think these are important confounders of the exposure and the stated outcomes.

It is not clear where the introduction ends and where the Study methodology starts

Study area: Under this section, the authors do not provide a justification for the choice of the hospital. The focus, apart from the population of Bikaner, should also be on what proportion of pregnant women use the facility for ANC and at what gestation they usually present. If 90% of pregnant women use the facility, the implications of the findings are completely different from when only 30% do. I think these are more relevant for the study than the TFR and CBR.

The study design: This must explicitly state that it is a cohort study.

Sample size: if the outcomes of the study are not explicitly cited, how are the sample sizes being considered? Also, to say "fairly large size" of study is very subjective. How many women did Ciborowski et al collect those 770 samples from? Was this a study with a "fairly large size" according to the authors? Same questions for the study by Maitre et al. the last sentence on page 8 of 29 is not clear. Was the sentence about the last two citations (63, 64) or it includes all the others (58-62)? I think the authors have not estimated the sample size. They need to contact a statistician to use valid assumptions for sample size estimations. Proposing a sample size to be "larger than the studies conducted to date" is simply not scientific. Even if the 1000 sample size is based on pragmatic considerations, the statistician can assess the power to be retained for various outcomes that will be explored in the study based on scientific assumptions. This has been attempted on L22 of pg 9 of 29 but again the authors mentioned cross-sectional study and did not have any assumptions about prevalence of outcomes.

Recruitment: With all these 9 criteria, I do not think the authors can get 2500 women out of the 6000 they aim to screen. First pregnancies are likely to have PIH or pre-eclampsia and how many women in rural India go to ANC clinic at 10 weeks of gestation? I think the authors have not provided a justification for all these stringent criteria for eligibility. The last paragraph under this section addresses too many issues and will need to be clearer.

Assessment procedures: The first paragraph mixes past and present continuous tenses. If the authors maintain the same sequence, it will improve comprehension?

o When are the questionnaires and the measurements on the pregnant woman being taken? The table 1 suggests that these are being taken in all three trimesters but I think there will be no need collecting some of these in all three trimesters.

o A clearer description of the timing of collection that will explain, for example, that when the participant's variables are not obtained in the first trimester, then they will be collected in the next visit. In that case, when women are not available in one trimester, when will the samples be collected. It needs to be clear what windows period are the sample collection taking place. If a participant is to have 1st trimester sample collected at 9 weeks, until when do they not collect the 1st trimester sample and move to the second?

	o I think the authors need to take a re-look at the procedures for taking the anthropometric measurements. I stand to be corrected but I do not think the MUAC is measured at the point of maximum diameter as espoused in the following “Mid-upper arm circumference is being measured as the maximum circumference of the upper arm taken horizontally where the biceps are most developed/widest (at the middle of the biceps)” o Are the Sphygmomanometers being used for the BP measurement calibrated for hyperdynamic circulation in pregnancy? If so, this must be stated clearly in the protocol. Otherwise, there is the likelihood of biases. Also, the Omron is a sphyg and so calibrating it with a sphyg is not clear. Again, taking BP measurements 2 mins apart may be ok but what is done when women are found to have “white coat hypertension” which is an apparent increase in the BP because of the fear of the machine. This usually resolves with re-assurance and repeat measurements after about 30 minutes. Among rural women, this might be significant. [ ] Estimation of gestational age: I recommend that the authors review the following statement: “If the difference between the CRL and LMP estimates is longer than ≤ 7 days, LMP is being considered valid and taken as the true biological date. In women with a LMP and CRL difference of >7 days, CRL is being considered valid for estimating the gestational age.” This is research and objective and standardized methods should be used to assign GAs. The standard has been early pregnancy ultrasound and so this must be adhered to so that the precision will be consistent. [ ] Biological sample collection: It is not clear what will be done with each sample. Does this sentence “The layer of WBC after separating plasma will be stored in an EDTA tube for future extraction of DNA for genetic studies” on line 27-29 is not clear. Describe what will happen to get the WBC layer and where that will be because the blood from the EDTA tube separates into plasma, buffy coat (WBCs) and RBCs as you did for the cord blood sample. Please by being systematic, readers can follow the methods. [ ] Data analyses: It may be useful for this section to be written such that it will be organised according to objectives of the analyses. Using two factors as exposures as well as outcomes as illustrated on line 43-44 may be confusing. [ ] References: I do not think the study requires 113 references. I suggest that the authors select the relevant literature to support what they want to do and not everything related to the subject.
--	---

VERSION 1 – AUTHOR RESPONSE

Reviewer: 1

Reviewer Name: Xiaobin Wang

Institution and Country: Johns Hopkins University, USA

Please state any competing interests or state 'None declared': None

Please leave your comments for the authors below

Summary:

This protocol describes a prospective study aiming to recruit and follow about 1,000 mother-offspring pairs to examine the association of maternal metabolites at each trimester with fetal metabolites at birth as well as maternal gestational weight gain and fetal growth and newborn anthropometric traits at birth.

Overall, the protocol is well-written. The scientific premise of this study is strong, and the research questions are important. The study design is appropriate for stated objectives. Data collection procedures and quality assurance are described in detail.

Response: We would like to thank the reviewer for providing valuable suggestions.

Specific comments: Study participants

Comment: This study is being conducted in a hospital located in Bikaner district (66% rural population), Rajasthan, India. It would be helpful to provide information on the socio-demographic characteristics of the pregnant mothers attending this hospital, including educational levels.

Response: We have added the socio-demographic and educational status of pregnant women visiting the hospital in "Study area" section.

Comment: Ethics: Please clarify if all the study participants have adequate literacy level to give written informed consent?

Response: Approximately 20% of the study participants are illiterate. Apart from the participants, we also take consent of the family members as witnesses in whose presence the study procedures are explained and informed consent is taken from the participants. It is unlikely that both the participant and the attendant are illiterate.

Minor comments:

1. Estimation of Gestational Age: On page 14 of 29, line 13, "is longer than ≤ 7 days", looks like a typo?

Response: We have now corrected the sentence.

2. On page 12 of 29, line 17, "being calibrated thrice a week", should be "twice"

Response: We have now changed the calibration into twice a week

Reviewer: 2

Reviewer Name: Alexander Manu

Institution and Country: Liverpool School of Tropical Medicine, UK

Please state any competing interests or state 'None declared': None declared

General comments: This is a relevant study that has substantial potential to contribute to a better understanding of the biological mechanisms underlying various adverse pregnancy and birth outcomes. The study of the metabolomics is becoming increasingly popular in global science and the

findings may have very wide applicability in low- and middle-income settings around the globe, where majority of these adverse pregnancy and birth outcomes occur.

Response: We would like to thank the reviewer for providing very helpful comments in detail that has now considerably improved the content of the manuscript.

However, these are my comments on the protocol

1. The order of presentation of the study procedures does not make easy reading. For instance, description of the quantitative analysis of the metabolomics is made before how and what samples are being collected and processed. Data collection is described before stating that the gestational age will be confirmed with ultrasound.

Response: The order of presentation has now been modified: ultrasound, estimation of gestational age, questionnaire, anthropometry, blood pressure, biological samples, laboratory tests, metabolomics, etc.

2. The manuscript could be made more precise and concise. There are too many repetitions and lots of material that do not add to the value of the research. It may be the reason why there are 113 references. Many of these can be taken out without affecting the relevance and quality of the study.

Response: We have revised the manuscript and added more information but the number of references are increased to 117.

3. The introduction does not make a case for the current gaps in knowledge which this study seeks to address

Response: The last sentence of the introduction clearly states (before the section on Metabolomic changes in pregnancy) that such studies are few in number. Moreover, three sub-sections before the section on "GaRBH cohort" have comprehensively covered the gaps in current knowledge.

4. There are some methodological and procedural imperatives that will need to be looked at carefully. For instance, the procedure for measuring MUAC; the decision to use Omron sphygms without stating whether they are calibrated for use in pregnancy; the choice of LMP over ultrasound for early pregnancies where the two differ; etc.

Response: The procedure for measuring MUAC has now been revised. The Omron blood pressure monitor is validated for pregnant women and the reference for the same has now been provided. It is acceptable to use LMP over ultrasound in early pregnancy for gestational age estimation if the difference between the two is ≤ 7 days (as used in INTERGROWTH study).

5. Sentence construction is very complex and sometimes the use of different tenses makes reading a little difficult.

Response: We have tried to simplify the sentence construction and made it more readable. The two different tenses are used in the manuscript purposely. Present tense is used for the study procedures that have already been started and are ongoing. Future tense is used for procedures which are yet to start (for example metabolomics, statistical analyses, infant anthropometry and body composition).

Comment: Title of the study: The title of the study may be slightly misleading. It is not clear what gestational route to healthy birth means. Every birth follows the gestational route irrespective of the outcomes – healthy or unhealthy birth. The study does not state that it is seeking to understand how to improve the gestational route in order to ensure a healthy birth. Moreover, I will suggest that the authors clearly indicate that this is the "protocol for a prospective cohort study" rather than "prospective cohort protocol". In my estimation, the two are not the same.

Response: We have modified the title “Gestational Route to Healthy Birth (GaRBH): protocol for an Indian prospective cohort study”. The term “gestational route to healthy word” was used to name the cohort because we are trying to understand the intrauterine influences (through the levels of metabolites) on foetal and birth outcomes. Moreover, it is an observational study and we have no plan of intervening in intrauterine environment for the improvement in gestational route which is only possible after the intra-uterine mechanisms clearly understood.

Abstract

Comment: In the abstract, the authors have made the for how important this study could potentially be. However, the import of a protocol is to state how the study is going to be done. There is nothing on what biological samples (blood or urine) are going to be collected, when and how these are going to be processed to ensure that the research questions can be answered.

Response: As per the suggestion of the reviewer, we have added the details of the biological samples in the abstract given the constraint in word limit.

Comment: Also, in the introduction the aim of the study is not clear as to whether the GWG is thought to depend on the variation in the level of the metabolites or the level of the metabolites will depend on GWG when the authors used the phrase “...in relation to GWG and its influence on...”, whereas in the methods, the analysis seems to be clearly looking at

“the role of gestational weight in influencing metabolite levels in each trimester and the association of maternal levels of metabolites with foetal growth, offspring’s weight and body composition at birth”

Even in the latter, the word “influencing” is scientifically unclear. Are they measuring association between GWG and metabolite levels and how these are, in turn, related to fetal growth and newborn anthropometric measurements? This needs to be clear.

Response: We have now clarified this in GaRBH cohort section. Our first objective has two components i.e. 1) to study the changes in levels of metabolites during the course of the pregnancy and 2) to examine the association of maternal gestational weight in each trimester with levels of metabolites in subsequent trimesters, and, to examine the association of levels of metabolites with GWG in subsequent trimesters.

Comment: Study strengths and limitations

What do the authors mean by “collecting objective measurements on pregnant women...”?

So far, the study objectives seem to suggest that the metabolites will be assessed in the mother in relation to the “influence” of their levels on fetal growth. Are the authors suggesting, in bullet 3, that metabolite levels in newborns will also be assayed?

Response: We have modified the 2nd point as

- The study is collecting biological measurements on pregnant women during each trimester (1st, 2nd and 3rd trimester) and their offspring at the time of birth.
- Yes, we will assay the metabolites in newborns using cord blood to meet the 3rd objective. This is also now described in Quantitative metabolomics section of the manuscript.

Comment: In bullet 4, the authors are exaggerating the strength of the study by claiming that most Indian cohorts are NOT from rural areas. Meanwhile they do not mention the obvious selection bias inherent in the fact that the study is entirely hospital-based. In the rural population, this is significant since many do not even have access to hospitals in order to be part of this study. Either the very ill mothers or the (relatively) richer population who can afford will end up being in hospital and hence will be recruited. This is a limitation which the authors need to mention.

Response: The study is based on government hospital (i.e. PBM, Bikaner), with nearly 1000 deliveries per month, which is accessible to all (including economically weaker sections) due to its free of cost health services. As per our screening of 1000 women to date, over 40% of them are coming from the villages. Moreover, due to various government initiatives, rural women are now persuaded to attend anti-natal clinics and deliver in hospitals as they are given incentives for medicines, diagnostics, transportation and institutional deliveries. Therefore, distribution of pregnant women visiting the antenatal care clinic of PBM hospital is not systematically biased.

Introduction

Comment: There are repetitions in the text that when avoided could have made the manuscript much shorter. For instance, pg5L5 and pg5L13 are both talking about the essence of the metabolic changes in pregnancy. The first paragraph therefore needs revision so that it becomes a bit more precise and concise.

Response: We have now removed the sentence on pg5L13.

Comment: The authors, on pg5L16, cited a study to support their claim on how metabolic changes promote fetal development but the said study found that increased FFAs only lowered maternal blood pressure and produced “more favourable metabolic phenotype for the mother”. The latter is quite vague and there was nothing about fetal development.

Response: This quoted statement in previous version was preceded by “Pregnancy is dominated by metabolic changes to meet the needs of foetal growth and development whilst protecting the health of the mother”. Therefore, it is about protecting mothers and not about foetal development”. In revised version we have removed the quoted sentence but kept the study as an example.

Comment: On pg5 paragraph2, it is not clear what the authors mean by system-wide understanding of metabolism. Is it “system-wide approach to understanding human metabolome”?

On L37 of para2, did you mean cells, tissues and organs?

Response: System-wide means across the biological systems of an organism. We have now clarified it as “biological system-wide” in revised manuscript. Metabolome can be studied at the level of cells, tissues and an organism (therefore, it is organism).

Comment: The last sentence of para2 does not read well. Am I right in thinking that the authors wanted to say that “only few longitudinal studies have quantitatively estimated the changes in the level of metabolomes at the different stages of pregnancy”? If so, the current construction needs revision.

Response: We have revised the sentence.

Comment: The two paragraphs under “Metabolomic changes in pregnancy” seem to contradict the last sentence just before that because whilst the authors claim studies quantifying metabolomics are few, in the 2 paragraphs alone, they have cited about 10 different studies. Could the authors also provide a chronological account of the cited changes during pregnancy from the first to the third trimester so that the picture becomes clearer to the reader?

Response: The studies cited under “Metabolomic changes in pregnancy” section are not longitudinal studies. Therefore, two paragraphs are not contradictory. In the 2nd paragraph, we have now added the trimesters studied by the cited research groups in order to clarify the non-longitudinal nature of the cited studies. Overall the number of metabolomic studies across the length of the pregnancy are few in number.

Comment: The section on “Relationships between maternal gestational weight gain and newborn health” has many very good points but will need to be re-written for the message to be clearer. As it is now, it reads as though the authors are presenting facts as they found them without guiding the reader to a clear message. It requires a bit of synthesis and focus.

Response: We have now revised this section for making it clearer.

Comment: In the section on “Maternal metabolic associations with newborn phenotypes”, again, the authors are clear with the message. The 3rd sentence on the Mendelian randomisation study could have stated the main findings i.e. causal associations between maternal BMI, blood glucose levels in pregnancy and infant birthweight. They could then have gone on to provide the exact quantification of the association and indicate statistical significance of the association. Also there was the mention of 10 metabolites but no identities of these metabolites are provided leaving the reader wondering what they might be. The authors on pg 7 of 29 L26 re cfRNA also refer to “maternal gestational age” when I think they were referring to the gestational age of a pregnancy.

Response: We have added the statistical significance and confidence interval of the cited mendelian randomization study and also added the names of metabolites associated with foetal growth. The term “maternal gestational age” has been changed to gestational age of a pregnancy.

Comment: Under the GaRBH cohort, I think you missed out “study” in the heading because you were not describing the cohort but the purpose of the study. Meanwhile, there is no clear rationale for the study because the previous paragraphs in the introduction have not clearly identified the current gaps in knowledge that the authors aim to address with this study. The question the reader will ask is “What is not known about this topic?”.

The research questions, therefore, seem to have no rationale.

Response: We have now changed it to “GaRBH Cohort Study”. We have also added more information on research gaps in the introduction section and sub-sections.

Comment: Study Objectives

The first objective starts with an assumption that the changes are going to be different in the different stages of pregnancy. This could be a subject of investigation under GaRBH. It could have been restated as “to determine whether the quantitative levels of metabolites differ at different stages of pregnancy and whether the trajectories are associated with maternal GWG”

The second objective should clearly state indicate that the associations will be measured independent of maternal pre-pregnancy BMI and paternal phenotypic characteristics. I think these are important confounders of the exposure and the stated outcomes.

It is not clear where the introduction ends and where the Study methodology starts

Response: Our first objective has two components i.e.

1) To study the changes in levels of metabolites during the course of the pregnancy, and, 2) To examine the association of maternal gestational weight each trimester with levels of metabolites in subsequent trimesters, and, to examine the association of levels of metabolites with GWG in subsequent trimesters.

We have now restructured the 2nd objective as per your valuable suggestions.

Comment: Study area: Under this section, the authors do not provide a justification for the choice of the hospital. The focus, apart from the population of Bikaner, should also be on what proportion of pregnant women use the facility for ANC and at what gestation they usually present. If 90% of pregnant women use the facility, the implications of the findings are completely different from when only 30% do. I think these are more relevant for the study than the TFR and CBR.

Response: The chosen hospital is the government facility which ensures accessibility for all due to its affordable nature and it is the largest hospital in Bikaner and caters to the majority of the pregnant women in the district. We have added this information in study area section of the revised manuscript. In India, it is usual that the pregnant women report to the ANC in the 2nd trimester but we are recruiting only those women who report before 13 weeks of gestation.

Comment: The study design: This must explicitly state that it is a cohort study.

Response: Cohort study is now explicitly mentioned in the study design

Sample size: if the outcomes of the study are not explicitly cited, how are the sample sizes being considered? Also, to say “fairly large size” of study is very subjective. How many women did Ciborowski et al collect those 770 samples from? Was this a study with a “fairly large size” according to the authors? Same questions for the study by Maitre et al. the last sentence on page 8 of 29 is not clear. Was the sentence about the last two citations (63, 64) or it includes all the others (58-62)? I think the authors have not estimated the sample size. They need to contact a statistician to use valid assumptions for sample size estimations. Proposing a sample size to be “larger than the studies conducted to date” is simply not scientific. Even if the 1000 sample size is based on pragmatic considerations, the statistician can assess the power to be retained for various outcomes that will be explored in the study based on scientific assumptions. This has been attempted on L22 of pg 9 of 29 but again the authors mentioned crosssectional study and did not have any assumptions about prevalence of outcomes.

Response: The meaning of “large” is relative to the majority of studies which are of limited sample size i.e. 30 to 60 samples. The last sentence that “the sample size must be larger than most of the informative studies” is for all the studies (Reference-58 to 64). A recent study, has included 80 pregnancies (28 pregnancies of normal foetal weight and 52 pregnancies of foetal smallness) and found significant differences in the level of metabolites (Miranda et al. 2018). Even the largest study on 4212 participants for studying the predictive metabolomic biomarker for placental complications included limited sample of preeclampsia (n=134), foetal growth restriction (n=162) and controls (n=259) (Gong et al. 2018). The outcome considered in this study are normal quantitative traits, like, levels of metabolites, gestational weight, foetal growth and anthropometry at birth, and certainly not for diseases. We have estimated power in terms of the amount of variation explained in levels of metabolites and foetal growth where full linear regression model (hypothesized model) was compared with null model.

References

- Miranda J, Simões RV, Paules C, Cañueto D, Pardo-Cea MA, García-Martín ML, Crovetto F, Fuertes-Martin R, Domenech M, Gómez-Roig MD, Eixarch E, Estruch R, Hansson SR, Amigó N, Cañellas N, Crispi F, Gratacós E. Metabolic profiling and targeted lipidomics reveals a disturbed lipid profile in mothers and fetuses with intrauterine growth restriction. *Sci Rep.* 2018 Sep 11;8(1):13614.
- Gong S, Sovio U, Aye IL, Gaccioli F, Dopierala J, Johnson MD, Wood AM, Cook E, Jenkins B, Koulman A, Casero RA Jr, Constância M, Charnock-Jones DS, Smith GC. Placental polyamine metabolism differs by fetal sex, fetal growth restriction, and preeclampsia. *JCI Insight.* 2018 Jul 12;3(13). pii: 120723.

Comment: Recruitment: With all these 9 criteria, I do not think the authors can get 2500 women out of the 6000 they aim to screen. First pregnancies are likely to have PIH or preeclampsia and how many women in rural India go to ANC clinic at 10 weeks of gestation? I think the authors have not provided a justification for all these stringent criteria for eligibility. The last paragraph under this section addresses too many issues and will need to be clearer.

Response: We are excluding women only if they report PIH in any of the previous pregnancies. Primigravida women once recruited in the study will not be excluded on the basis of PIH or preeclampsia. We agree that it is a challenge to recruit 2500 healthy pregnant women out of 6000 screened women, however, all these criteria are essential to recruit only normal healthy pregnant women. We will revisit our eligibility criteria periodically to achieve the target sample size.

Assessment procedures: The first paragraph mixes past and present continuous tenses. If the authors maintain the same sequence, it will improve comprehension? o When are the questionnaires and the measurements on the pregnant woman being taken? The table 1 suggests that these are being taken in all three trimesters but I think there will be no need collecting some of these in all three trimesters.

Response: The two different tenses are used in the manuscript purposely. Present tense is used for the study procedures that have already been started and are ongoing. Future tense is used for procedures which are yet to start (for example metabolomics, statistical analyses, infant anthropometry and body composition).

Comment: A clearer description of the timing of collection that will explain, for example, that when the participant's variables are not obtained in the first trimester, then they will be collected in the next visit. In that case, when women are not available in one trimester, when will the samples be collected. It needs to be clear what windows period are the sample collection taking place. If a participant is to have 1st trimester sample collected at 9 weeks, until when do they not collect the 1st trimester sample and move to the second?

Response: We will exclude the women whose data for each trimester is not available which is why the first trimester count of 2500 may finally reduce to 1000 women-offspring pairs. We have clearly mentioned the timelines of data collection in Table-1 and Figure-2. Measures like ultrasound, anthropometry and blood are collected on the same day, and whenever the participant's variables related to questionnaire are not obtained on the same day in that case they are rescheduled within next three days.

Comment: I think the authors need to take a re-look at the procedures for taking the anthropometric measurements. I stand to be corrected but I do not think the MUAC is measures at the point of maximum diameter as espoused in the following "Mid-upper arm circumference is being measured as the maximum circumference of the upper arm taken horizontally where the biceps are most developed/widest (at the middle of the biceps)"

Response: MUAC is measured at the at the mid-point between the tip of the shoulder and the tip of the elbow (acromion and olecranon process, respectively) taken horizontally (at the middle of the biceps). We have now modified the statement in adult anthropometry section.

Comment: Are the Sphygmomanometers being used for the BP measurement calibrated for hyperdynamic circulation in pregnancy? If so, this must be stated clearly in the protocol. Otherwise, there is the likelihood of biases. Also, the Omron is a sphyg and so calibrating it with a sphyg is not clear. Again, taking BP measurements 2 mins apart may be ok but what is done when women are found to have "white coat hypertension" which is an apparent increase in the BP because of the fear of the machine. This usually resolves with re-assurance and repeat measurements after about 30 minutes. Among rural women, this might be significant.

Response: We have used Omron HBP-1300 which is validated in pregnant women (Abbud et al. 2018) and we are calibrating the Omron machine with sphygmomanometer thrice a week. The sphygmomanometer used for calibration are not calibrated for hyper-circulation. Our field staff is trained to establish rapport with the participants to avoid situations like “white coat hypertension” in our study population.

Comment: Estimation of gestational age: I recommend that the authors review the following statement: “If the difference between the CRL and LMP estimates is longer than ≤ 7 days, LMP is being considered valid and taken as the true biological date. In women with a LMP and CRL difference of >7 days, CRL is being considered valid for estimating the gestational age.” This is research and objective and standardized methods should be used to assign GAs. The standard has been early pregnancy ultrasound and so this must be adhered to so that the precision will be consistent.

Response: INTERGROWTH study has recommended that both LMP and CRL should be used for foetal growth monitoring. If the difference between the CRL and LMP is ≤ 7 days, the LMP will be considered valid and taken as the true biological date (Villar et al. 2013). One of the limitations of ultrasound based gestational age estimation is that for a given CRL value all the foetuses will have the same gestational age, because it excludes any biological variability (Villar et al. 2013).

Comment: Biological sample collection: It is not clear what will be done with each sample. Does this sentence “The layer of WBC after separating plasma will be stored in an EDTA tube for future extraction of DNA for genetic studies” on line 27-29 is not clear. Describe what will happen to get the WBC layer and where that will be because the blood from the EDTA tube separates into plasma, buffy coat (WBCs) and RBCs as you did for the cord blood sample. Please by being systematic, readers can follow the methods.

Response: We have made it clear in the manuscript that the EDTA tube is centrifuged at 3500 rpm for the separation of WBCs, plasma and RBCs.

Comment: Data analyses: It may be useful for this section to be written such that it will be organised according to objectives of the analyses. Using two factors as exposures as well as outcomes as illustrated on line 43-44 may be confusing.

Response: It is organized as per our objectives. We need to analyse GWG and levels of metabolites while considering them exposure and outcomes both for which we have added another sentence in data analysis: “We will also analyse the associations of level of metabolites from first to third trimester with the trajectories of gestational weight gain”.

Comment: References: I do not think the study requires 113 references. I suggest that the authors select the relevant literature to support what they want to do and not everything related to the subject.

Response: The number of references are now 117 due to additional information provided after responding to reviewers comments.

VERSION 2 – REVIEW

REVIEWER	Xiaobin Wang Johns Hopkins University, Baltimore, USA
REVIEW RETURNED	22-Oct-2018

GENERAL COMMENTS

The revised manuscript has adequately addressed my comments.
I have no further question.